# Body Image Dissatisfaction as a Risk Factor for Postpartum Depression

**DOI:** 10.3390/medicina58060752

**Published:** 2022-05-31

**Authors:** Francisco Javier Riesco-González, Irene Antúnez-Calvente, Juana María Vázquez-Lara, Luciano Rodríguez-Díaz, Rocío Palomo-Gómez, Juan Gómez-Salgado, Juan Jesús García-Iglesias, Tesifón Parrón-Carreño, Francisco Javier Fernández-Carrasco

**Affiliations:** 1Department of Obstetrics, Punta de Europa Hospital, 11207 Algeciras, Spain; javieriesco75@gmail.com (F.J.R.-G.); irene_antunez@hotmail.es (I.A.-C.); juana.vazquez@uca.es (J.M.V.-L.); fjavier.fernandez@uca.es (F.J.F.-C.); 2Nursing and Physiotherapy Department, Faculty of Nursing, University of Cádiz, 11207 Algeciras, Spain; 3School of Health Sciences, University of Granada, 51003 Ceuta, Spain; lucianord@ugr.es; 4Department of Obstetrics, Línea de la Concepción Hospital, 11300 Línea de la Concepción, Spain; rociopalomogomez@hotmail.es; 5Department of Sociology, Social Work and Public Health, Faculty of Labour Sciences, University of Huelva, 21007 Huelva, Spain; juanjesus.garcia@dstso.uhu.es; 6Safety and Health Postgraduate Programme, University of Espíritu Santo, Guayaquil 092301, Ecuador; 7Territorial Delegation of Equality, Health and Social Policies, Health Delegation of Almeria, Government of Andalusia, 04003 Almeria, Spain; tesifon.parron@juntadeandalucia.es

**Keywords:** postpartum depression, body image, puerperium, risk factors

## Abstract

*Background and Objectives*: Pregnancy and postpartum are periods that imply numerous physical and psychological changes that could lead to mental health consequences. The aim of the present study is to identify whether women who had body image dissatisfaction had a higher incidence of postpartum depression 6 months after birth than women who did not have body image dissatisfaction. *Materials and Methods*: A descriptive cross-sectional study was designed with a sample of 450 women from two hospitals in Andalusia. Quantitative variables were age and scores on the Edinburgh Postnatal Depression Scale (EPDS) and the Body Shape Questionnaire (BSQ) for body image dissatisfaction. The qualitative variables used were marital status, self-perceived health status, diet or physical exercise, type of delivery, and others. *Results*: Body dissatisfaction was positively correlated with the symptomatology of postpartum depression. Thus, for each point increased in body dissatisfaction, the occurrence of depression also increased. There was a relationship between the study variables, so women who were more dissatisfied with their body image were more frequently depressed. *Conclusions*: In conclusion, it can be established that postpartum depression seems to be related to the presence of poor body image.

## 1. Introduction

Major depressive disorder is the leading cause of illness in developed countries [1]. This disorder, despite belonging to the field of mental health, is frequently treated in the primary care (PC) setting. In this sense, most people who present depressive symptoms do not always seek care for their psychological symptoms, but usually attend for somatic problems such as headaches, myalgia, or gastralgia [2]. Despite the wide availability of antidepressant medication and psychotherapy, studies show that only 14–56% of these individuals receive adequate treatment [1].

Postpartum depression (PPD) describes depressive symptoms in women during the first postpartum year. It is characterised by negative mood, decreased activity and social behaviour, low energy and motivation, reduced spontaneous experiences, ideas, and pleasurable activities (anhedonia), and is associated with an increase in self-harm and even suicidal behaviour [3].

Although PPD affects 11–20% of women who have given birth [4], and has quite negative consequences for both the mother and the newborn, it remains undertreated and underdiagnosed, and is most often misinterpreted [5]. This is because women do not always show symptoms of PPD when they attend a consultation, and often even think that the cause of their discomfort is something common in that period, related to lack of sleep or other factors that they disregard as irrelevant [5]. Only 11% of women express their depressive symptoms to their physician and many of them minimise or even deny possible depressive disorders for fear of being judged on their ability to be a mother [6]. However, PPD is becoming a public health problem, associated with increased maternal mortality [3]. Currently, in the European healthcare system, women have access to clinical and, to a lesser extent, mental health care. This may explain why rates of mental health problems related to pregnancy are so high [7].

Previous studies have focused mainly on PPD [8], without taking into account other important mental health variables such as anxiety [9] or changes in the body as perceived by the woman during pregnancy [10]. In this sense, 12.7% of women are diagnosed with a major depressive disorder during pregnancy, which is why it would be advisable to have a check-up in each trimester of pregnancy, as advised by the American College of Obstetricians and Gynaecologists (ACOG) [11], in order to detect depression prior to delivery. According to the cognitive theory, depression is based on attitudes that develop from previous experiences, so when it occurs it can be rigid and maladaptive [12]. Depressed people often have negative evaluations of their own body, which may lead to body dissatisfaction [13]. Conversely, other studies have assessed the reverse direction, i.e., body dissatisfaction as a cause of depression [14].

On the other hand, body image is defined as the way a person perceives him/herself, as well as the perception of how others see us [15]. During pregnancy, considered a rather complex process, women experience changes both physically (weight gain, changes in skin pigmentation, hyperflexibility of joints, changes in the cardiovascular, digestive, and respiratory systems) and psychologically (risk of low self-esteem, demotivation, anxiety, stress) [16]. Because of these circumstances, women may pay more attention to these transformations occurring in their own body, which could lead to a negative perception of it, and therefore to an increase in dissatisfaction with their body image [17].

The problem of body image dissatisfaction is a multidisciplinary component that can have very negative consequences on personality, behaviour, and social relationships. This may be due, among other factors, to the communicative capacity of our body in social relationships [18].

In Roomruangwong’s study [19] a positive relationship between anxiety and body dissatisfaction was reported. Others such as Chan et al. [9] have linked anxiety to eating disorders during pregnancy. This background may indicate that pregnancy is a time when women present themselves as highly sensitive, in terms of both their body and emotions, with body dissatisfaction appearing as a common factor. Therefore, women with body dissatisfaction may show a greater tendency to experience negative processes (psychological and physical) that could become pathological.

Thus, most studies have focused on the levels of both factors individually (body dissatisfaction and depression), but these two variables are highly comorbid with other factors that have not been taken into account [19]. These could be demographic (single-parent family), histories of past addictions (alcoholism), or psychosocial factors (low self-esteem, low support, unwanted pregnancy, family violence, or victim of bullying [20].

The main objective of the present study was to relate body image dissatisfaction and postpartum depression in postpartum women who were treated in a hospital in Andalusia, belonging to the Andalusian Health Service, which integrates the public health centres of the region.

## 2. Materials and Methods

### 2.1. Study Design

A descriptive cross-sectional study was designed using questionnaires. The study design was based on the following research question: Is there a relationship between body image (dis)satisfaction and the likelihood of developing postpartum depression in postpartum women? According to the PECO format, the (P)opulation is postpartum women, the (E)xposition are women who are dissatisfied with their body image (women with Body Shape Questionnaire (BSQ) scores above 20.6), (C)ontrol refers to the women who are satisfied with their body image (women with scores at or below 20.6 on the BSQ), and (O)utcomes are scores on the Edinburgh Postnatal Depression Scale (EPDS). The covariates taken into account were: marital status, employment status, socio-economic status, educational level, having a mental health history, being on a weight loss diet, physical activity, child feeding method, having support from others, self-perceived health status, and the type of delivery.

### 2.2. Population and Sample

The sample was chosen from births occurring in the study hospital from 1 July 2020 to 30 June 2021. In this period, the total number of births in the centre was 970.

The sample size was calculated with the Sample Size Calculator programme, starting from a total population of 970 births, applying a confidence level of 95%, and with a maximum error of 5%, it was obtained that a sample of at least 276 subjects would be necessary. The sample that was achieved was larger and consisted of a total of 449 subjects.

The randomisation process was carried out systematically, randomly selecting all women who visited the Hospital Punta de Europa for a follow-up of their pregnancy, met the inclusion criteria, and wished to participate in the study voluntarily.

The inclusion criteria established were: women who had given birth 6 months prior to participating in the study; eutocic, dystocic, and caesarean deliveries in which the delivery had taken place without peripartum pathologies; the newborns had been born healthy, were of normal weight and without any neonatal pathology, and did not require more time in hospital than usual.

The exclusion criterion was the existence of a language barrier in terms of the Spanish language (the patient could not write, read, or speak in Spanish). In many cases, the companion or direct relative did understand Spanish, but even so, this requirement was maintained as an exclusion criterion. Maternal age above 44 years was also established as an exclusion criterion, as it is estimated that above this age perimenopausal symptoms may appear, which are not the subject of this study.

### 2.3. Instruments

Three questionnaires were used for data collection: one for socio-demographic data and two validated questionnaires: the Edinburgh Postnatal Depression Scale (EPDS) and the Body Shape Questionnaire (BSQ).

The EPDS was designed for screening for PPD in primary care and contains 10 items with four possible response alternatives according to the severity of depressive symptoms felt by the women during the last week, also scored from 0 to 3. The total range is from 0 to 30. The authors propose a cut-off point of 10 to classify the depressive state [21]. For the Spanish population, a cut-off point of 10 was established, so that women who scored 11 or more were considered to have depressive symptoms. The reliability of the scale was tested by measuring the Cronbach’s alpha coefficient, which was 0.88 for our sample.

The BSQ or Body Shape Questionnaire by Cooper et al. [22] is a self-administered questionnaire that assesses body dissatisfaction, fear of becoming fat, feelings of low self-esteem due to physical appearance, desire to lose weight, and avoidance of situations where physical appearance might attract the attention of others. The abbreviated version of the questionnaire was previously used by Evans and Dolan [23] in 1993. The questionnaire consists of 14 items and is scored on a Likert scale of 1–6: ‘never’, ‘rarely’, ‘sometimes’, ‘often’, ‘very often’, and ‘always’. Higher values on the BSQ indicate greater body dissatisfaction. The score can range from 14 to 84 points. It is a questionnaire with high internal consistency and reliability; in our sample, the reliability of the scale was measured using Cronbach’s alpha coefficient with a value of 0.87.

In addition to the scores of the two tests described above, age and number of children were taken as quantitative variables. Qualitative variables were marital status, level of education, employment status, socio-economic status, family model, history of mental health, feeding of the newborn, support person or immediate family member, weight loss diet, physical activity, self-perception of health status, type of delivery, and whether stitches were used in the delivery.

### 2.4. Data Collection

Data were obtained between 01 January and 31 December 2021, from the retrospective patient management database of the public hospitals of Andalusia (Spain). This database is generic for the entire Andalusian public health system.

Once the patient was selected, a member of the research team would contact her by telephone, so as to be informed of the characteristics of the study and invited to participate. Once the patient gave her consent, a link was then sent to her via WhatsApp so that she could access and answer the questionnaire comfortably from her phone.

### 2.5. Data Analysis

The descriptive analysis of quantitative variables was carried out using measures of central tendency and dispersion. Categorical variables were described in absolute numbers and percentages. In the bivariate statistical analysis, to relate quantitative variables and after the Kolmogorov–Smirnov normality test, non-parametric tests were used: Mann–Whitney U test to compare two independent variables and Kruskal–Wallis test for more than two. Finally, a multivariate binary logistic regression model was developed to establish associations between the different independent variables and the dependent variable. Confidence intervals (CI) were obtained at 95% and a significance level of *p* < 0.05.

The statistical study was carried out with the IBM SPSS Statistics version 26.

## 3. Results

### 3.1. Descriptive Analysis

The mean age of the women was 32 years, with a standard deviation (SD) of 5.83, the minimum age being 18 years and the maximum being 44 years. The mean number of children in the families was 1.69, with a minimum of 1 and a maximum of 5. The mean score of the Edinburgh depression test was 8.65 and the mean score of the Body Shape Questionnaire was 20.6 (Table 1).

The descriptive analysis of the qualitative variables is shown in Table 2.

### 3.2. Correlational Analysis

The data for the body image dissatisfaction (BSQ) and depression (EPDS) variables did not follow a normal distribution. For this reason, non-parametric tests were used.

Body dissatisfaction was positively correlated with postpartum depression, so that for each point increased in body dissatisfaction, depression also increased (Table 3).

### 3.3. Comparisons of Means and Bivariate Analysis

#### 3.3.1. EPDS Test and BSQ Related to Marital Status

For the EPDS, the mean for single women was 8.92, 8.50 for married women, and for divorced women it was 3.33. Statistically significant differences were found (*p* = 0.03).

The post hoc test confirmed the significant differences between the divorced-married (*p* = 0.03) and divorced-single groups (*p* = 0.02).

For the BSQ, no statistically significant differences were found.

#### 3.3.2. EPDS Test and BSQ Related to Employment Status

According to employment status, statistically significant differences were found for both tests (EPDS and BSQ).

The mean EPDS score was 9.4 for unemployed women, 7.75 for working women, and 9.65 for women who were engaged in household care (*p* = 0.02). Women who were unemployed or engaged in household care had higher EPDS scores. The post hoc test found no statistically significant differences between the groups.

The mean on the BSQ was 22.52 for unemployed women, 19.25 for working women, and 21.45 for women engaged in household care (*p* = 0.01). The post hoc test confirmed statistically significant differences between the group of working women and the group of unemployed women (*p* = 0.01).

#### 3.3.3. EPDS Test and BSQ by Socio-Economic Status

According to the socio-economic status, statistically significant differences were found for both variables (EPDS and BSQ).

The mean on the EPDS was 10.89 for women with low income (<1000 euros/month), 8.25 for those with middle income (1000–2000 euros/month), 6.99 for those with middle-high income (2000–3000 euros/month), and 9.07 for those with high income (>3000 euros/month) (*p* = 0.001). The post hoc test confirmed statistically significant differences between the high-middle income group and the low-income group (*p* = 0.001), and between the middle-income group and the low-income group (*p* = 0.11).

The mean on the BSQ was EPDS was 20.46 for women with low income (<1000 euros/month), 21.74 for those with middle income (1000–2000 euros/month), 18.43 for those with middle-high income (2000–3000 euros/month), and 18.9 for those with high income (>3000 euros/month) (*p* = 0.04). The post hoc test confirmed statistically significant differences between the high-middle income group and the middle-income group (*p* = 0.03).

#### 3.3.4. EPDS Test and BSQ Related to Educational Level

When relating the mean of the tools (EPDS and BSQ) with the level of education, statistically significant differences were found for both questionnaires (EPDS and BSQ).

The mean on the EPDS was 3.2 for women with no studies, 9.01 for women with primary studies, 8.87 for women with intermediate studies (higher secondary studies or vocational training), and 8.32 for women with university or postgraduate studies (*p* = 0.03). The post hoc test confirmed statistically significant differences between the group of women with no studies and the group of women with university/postgraduate studies (*p* = 0.01); also, between the group of women with no studies and the group of women with intermediate studies (*p* = 0.009) and between the group of women with no studies and the group of women with primary studies (*p* = 0.004).

The mean on the BSQ was 11.8 for women with no studies, 20.93 for women with primary studies, 21.08 for women with intermediate studies (higher secondary studies or vocational training), and 20.11 for those with university or postgraduate studies (*p* = 0.007). The post hoc test confirmed statistically significant differences between the group of women with no studies and the group of women with university/postgraduate studies (*p* = 0.005); also, between the group of women with no studies and the group of women with intermediate studies (*p* = 0.004) and between the group of women with no studies and the group of women with primary studies (*p* = 0.003).

#### 3.3.5. EPDS Test and BSQ Related to Having a Mental Health History

The mean score on the EPDS test for people without mental health problems was 7.44 versus 13.42 for people who reported having mental health problems (*p* = 0.001). The mean score on the BSQ test for people who reported having had no mental health problems was 19.48, versus 24.97 for people who reported having had mental health problems (*p* = 0.001).

#### 3.3.6. EPDS Test and BSQ Related to Being on a Weight Loss Diet

The mean score on the EPDS test for women who were not on a weight loss diet was 8.28, while the mean score for dieters was 9.69 (*p* = 0.03). For the BSQ test, the mean score of the non-dieters was 19.02, while dieters had a mean score of 25.00 (*p* = 0.001).

#### 3.3.7. EPDS Test Related to Physical Activity

In the case of physical activity, statistically significant differences were found for depression.

The mean score on the EPDS was 9.18 for women who reported not performing any physical activity, 9.05 for women who were physically active once a week, 7.43 for those who were physically active 2–3 times a week, and 5.56 for women who were physically active daily (*p* = 0.01). The post hoc test showed no statistically significant differences between the groups.

#### 3.3.8. BSQ Related to Child Feeding Method

Statistically significant differences were found for body image dissatisfaction in relation to the way the child was fed.

The mean BSQ score for exclusively breastfeeding mothers was 20.60, for formula feeding mothers it was 21.43, and for mixed breastfeeding mothers, 17.49 (*p* = 0.02). The post hoc test showed statistically significant differences between the exclusive breastfeeding group and the formula feeding group (*p* = 0.01).

EPDS was not discussed in terms of method of infant feeding because no statistically significant differences were found for this variable.

#### 3.3.9. EPDS Test and BSQ Related to Having Support from Others

When comparing the depression and dissatisfaction with body image variables (EPDS and BSQ) with having support from others, statistically significant differences were found for both tests. Women who reported having no support had a higher score in the EPDS and the BSQ.

The mean for the EPDS was 15 for women who had no support, 7.85 for women whose main support was their partner, 10.77 for women who were supported by their mother or mother-in-law, and 11 for those who reported having support from someone other than their partner (*p* = 0.001). The post hoc test established statistically significant differences between the group with no support and the group whose main support was their partner (*p* = 0.019); also, between the group of women who felt supported by their partner and the group whose support was given by their mother or mother-in-law (*p* = 0.001).

The mean on the BSQ was 29 for women who had no support, 20.01 for women whose main support was their partner, 20.65 for women who were supported by their mother or mother-in-law, and 29.67 for those who reported having support from someone other than their partner (*p* = 0.001). The post hoc test established that the statistically significant differences were between the group that did not have any support and the group whose main support was their partner (*p* = 0.049); and between the group of women who felt supported by their partner and the group whose support was given by their mother or mother-in-law (*p* = 0.002).

#### 3.3.10. EPDS Test and BSQ Related to Subjective Health Status

When the variables EPDS and BSQ were compared with subjective health status, significant differences were found for both.

The mean for EPDS was 6.76 for women who considered their current health status to be good and 14.17 for those who felt their current health status was mediocre (*p* = 0.001). No women reported that their current health status was poor (*p* = 0.001).

#### 3.3.11. EPDS Test and BSQ Related to the Type of Delivery

When relating EPDS and BSQ variables to the type of delivery, statistically significant differences were found for both.

The mean EPDS score was 8.81 for women who had had a normal delivery, 6.48 for women who had an instrumental delivery, and 9.38 for women who had a caesarean section. The post hoc test showed statistically significant differences between the group of women who had had a normal delivery and the group of women who had had a caesarean section (*p* = 0.02), and also between the group of women who had had an instrumental delivery and the group of women who had had a caesarean section (*p* = 0.001).

The mean BSQ score was 20.68 for women who had had a normal delivery, 16.07 for women who had had an instrumental delivery, and 22.72 for women who had had a caesarean section. The post hoc test showed statistically significant differences between the group of women who had had a normal delivery and the group of women who had had a caesarean section (*p* = 0.003), and between the group of women who had had an instrumental delivery and the group of women who had had a caesarean section (*p* = 0.002).

### 3.4. Multivariate Analysis

In order to carry out the multivariate analysis, a binary logistic regression was performed (Table 4). In a first step, depression was taken as the dependent variable, and all the variables that showed statistical significance in the bivariate analysis were taken as independent variables. In order to be able to use depression as the dependent variable, it was necessary to recode it dichotomously. To do this, following the indications of the creators of the scale, scores equal to or greater than 10 were coded as people with depression, and scores below 10 were coded as people who did not have depression.

Age and income were identified as protective factors. As risk factors, single-parent family model, subjective self-perception of health, and having received stitches in childbirth were detected.

In a second step, body image dissatisfaction was taken as a dependent variable, and all variables that showed statistical significance in the bivariate analysis as independent variables. In order to use dissatisfaction with body image as the dependent variable, it was necessary to recode it dichotomously, so that values below the mean (20.6 points) were considered as people who were satisfied with their body image, and values above the mean were considered as people who were dissatisfied with their body image (Table 5).

Physical activity was identified as a protective factor, and dieting, caesarean delivery, and depression as risk factors.

## 4. Discussion

The importance of mental health during pregnancy and the puerperium has been sufficiently contrasted [24]. Depression [1], anxiety [25], and body dissatisfaction [10] are the most frequent problems. In the present study, the aim was to determine the possible relationship between body dissatisfaction and depression in a sample of women 6 months after childbirth.

Body dissatisfaction before and during pregnancy leads to higher levels of PPD in the postpartum period [9,19,26]. However, other studies have been found that regard the period of pregnancy as a protective factor for body dissatisfaction [27]. These studies suggest that pregnancy is a period of happiness, in which women downplay their physical change, feelings of inadequacy or despondency, which would result in a lesser impact of these changes on mental health-related aspects such as those studied here [27]. The present study results indicate that women with greater dissatisfaction with their body image also had higher levels of depression. This positive correlation has also been described in several studies [6,9,14].

The existing literature has documented associations between some pre-birth situations, such as marital status, and maternal depression during the first year [28,29,30,31]. Our results show that single women had higher scores on the EPDS than married women, but divorced women had less depression than the rest. However, these results should be interpreted with caution as the group of divorced women was very small (*n* = 6) and therefore this result should not be generalised. On the other hand, Akincigil et al. [32] studied the relationship between postpartum depression and marital status, although they found that being married or single was less important than the quality of the relationship. They concluded that, after controlling for relationship quality, single women were not more likely to be depressed than married or cohabiting women.

In the present study, statistically significant differences were found when employment status was related to postpartum depression. Women who were unemployed or engaged in home care had higher levels of depression than women who were working. These results are consistent with those published by Lewis et al. [33], who suggested that being employed is a protective factor for depression.

Regarding socio-economic status, studies confirm that low economic status is correlated with higher levels of depression [34]. These data were confirmed in the present study, and these women were also more dissatisfied with their body image.

With respect to the level of education of the participants, women with a lower level of education had less depression and less dissatisfaction with their body image than those with a higher level of education. These results are in agreement with the study by van der Zee-van den Berg et al. [35], who state that the higher the level of education, the higher the levels of depression and anxiety in the postpartum period. However, Matzumura et al. found the opposite, but this relationship proved to be very weak when regression models were fitted [36].

It is well documented that women with a history of anxiety and/or depression are more likely to develop PPD [37,38,39]. This has been confirmed in the present study. Something similar occurs in the case of body image dissatisfaction. The results are consistent with previous studies in finding that women with a history of anxiety and depression are also more dissatisfied with their body image [19].

Women gain weight with pregnancy, and after childbirth they are most concerned about losing this excess weight. It has been reported that being overweight is a reason for greater dissatisfaction with body image and more PPD [40]. In the present study, women who were on a weight loss diet had more PPD and were more dissatisfied with their body image and, ultimately, this is the reason why they started a weight loss diet. Another important aspect of weight loss is physical activity, and women who did not perform physical activity at all had higher scores on the EPDS. These results are consistent with other recent studies [40].

During pregnancy, fat accumulates to ensure foetal development and later breastfeeding. If a woman has been well nourished during pregnancy, her fat stores can provide about one third of the energy and essential fatty acids needed during the first three months of lactation [41]. In this study, women who did not breastfeed were more dissatisfied with their body image. This may be due to the fact that breastfeeding helps to restore the figure by eliminating excess fat that accumulated during pregnancy as well as being satisfying for the mother because of the increased bonding with the newborn [42].

Many studies have included variables such as body image, eating attitudes, or breastfeeding in their theoretical construct [43,44], but very few have taken into account the woman’s family or social support [45], and that having support during the postpartum period is very important, regardless of the relationship between the support person and the mother. In the present study, women who were supported by their partners had lower levels of PPD and less dissatisfaction with their body image, followed by those who were supported by their mothers. Women who reported lacking support had higher levels of depression and more dissatisfaction with their body image. These findings are consistent with results published in the research by Rodgers et al. [45], who studied the influence of partners on women’s body image and eating attitudes in the postpartum period.

The type of delivery also influences PPD. In the meta-analysis by Xu et al. [46], which included 27 articles with a total of 532,630 women, it was concluded that delivery by caesarean section increased the risk of PPD. These results are consistent with those of the present study, where it was observed that women with caesarean section had higher levels of postpartum depression and more dissatisfaction with their body image. However, some studies found no significant differences between PPD and the type of delivery [47,48].

The binary logistic regression model, taking depression as a dependent variable, concluded that age is a protective factor; the older the age, the lower the risk of depression. Income also behaved as a protective factor; the higher the income, the lower the risk of depression. With respect to the family model, the single-parent family had 5.7 times more risk of suffering from depression. As regards people who reported mediocre subjective health, they were 13.41 times more likely to suffer from depression. Finally, with respect to receiving stitches, women who were most at risk of suffering from depression, despite having had a vaginal birth, were those who had an episiotomy after a normal birth. In the study by Petrosyan et al. [49] it was observed that women under 25 years of age were more depressed when the delivery was by caesarean section. However, when the delivery was vaginal, there was no increase in depression levels. Many studies also indicate that families with a single-parent structure are more at risk for PPD [50,51,52].

In the other logistic regression model in which body image dissatisfaction was taken as the dependent variable, people who reported being on a weight loss diet were 4.71 times more likely to be dissatisfied with their body image. Moreover, physical activity was a protective factor for body image dissatisfaction, caesarean delivery implied a 1.89-fold risk for body image dissatisfaction, and depression was related to body image with an OR of 1.11. Women who were dissatisfied with their body image often decided to start a weight loss diet to try to improve their figure [53]. For this reason, many women decided to go on a weight loss diet and start physical activity with the dual aim of losing weight and improving their body image [54]. Consistent with this study results, some studies state that women who gave birth by caesarean section are more likely to be dissatisfied with their body image [54].

There are several limitations in the present study. One of them could be that the sample was taken from a single centre. Although the sample size was calculated to be representative in number, a sampling bias must not be disregarded due to the socio-cultural profile; the number of immigrants in this area due to its geographical location may not be comparable with the rest of the country, or even with the rest of the European population. In the study, the place of origin of the puerperae was not taken into account. Another limitation is based on the way data were collected. Data were self-reported by the patient, which could lead to forgetting bias and gift bias. It would be interesting for future research to expand the sample, not so much in number as in geographical dispersion. Last but not least, it should be noted that 20.3% of the sample reported a history of anxiety or depression before the birth. Thus, it is possible that what we are calling PPD is actually a continuation of the previous condition. However, this variable was included because there are studies that show that women with anxiety and depression prior to childbirth have a higher risk of developing PPD.

## 5. Conclusions

Scientific evidence supports that PPD is a disorder that depends on many variables. The results obtained in this study reveal that dissatisfaction with body image directly affects PPD, favouring its appearance; therefore, it is necessary to promote a positive attitude towards body image and provide women with the necessary tools to improve their body image in the postpartum period, thus reducing the risk of suffering PPD.

It is important from this point of view for pregnant women to be informed through their health care providers of everything related to the physiological and anatomical changes that occur during pregnancy and the postpartum period; this will allow them to gradually adapt to the transformations that their bodies will undergo, understanding why they are necessary and transitory and, in the end, natural. It would be interesting to expand research on this topic with an emphasis on testing the effectiveness of preventive measures for this health problem.

## 6. Ethical Aspects

This research had a favourable opinion from the Andalusian Biomedical Research Ethics Committee (code 1250-N-21). The general principles of the Declaration of Helsinki, updated in 2013 in Brazil, which list the ethical principles in research involving human subjects, were taken into account throughout the conduct of this study by the entire research team. In addition, the provisions of current Spanish legislation on biomedical research (Law 14/2007, of 3 July, on Biomedical Research and Law 41/2002, of 14 November, on patient autonomy and rights and obligations regarding clinical information and documentation) and on personal data protection were also followed.

## Figures and Tables

**Table 1 medicina-58-00752-t001:** Descriptive analysis of quantitative variables.

	N	Minimum	Maximum	Mean	Standard Deviation
Age	449	18	44	31.99	5.829
No of children in the family	449	1	5	1.69	0.760
TOTAL EPDS TEST	449	0	23	8.65	5.670
TOTAL BSQ TEST	449	8	44	20.60	8.9

**Table 2 medicina-58-00752-t002:** Descriptive analysis of qualitative variables.

Variable	Values	Frequency	Percentage
Marital status	Single	234	52.1
	Married	209	46.5
	Divorced	6	1.3
	Total	449	100
Level of studies	No studies	10	2.2
	Primary studies	134	29.8
	Higher secondary studies or vocational training	195	43.4
	University or postgraduate	110	24.5
	Total	449	100
Employment situation	Unemployed	99	22
	Working	223	49.7
	Household care	127	28.3
	Total	449	100
Economic level (income)	<1000 euros	102	22.7
	Between 1000 and 2000 euros	226	50.3
	Between 2000 and 3000 euros	91	20.3
	>3000 euros	30	6.7
	Total	449	100
Family model	Heteroparental	434	96.7
	Single-parent	15	3.3
	Total	449	100
History of mental health	Yes	91	20.3
	No	358	79.7
	Total	449	100
Newborn feeding	Artificial	222	49.4
	Exclusive breastfeeding	159	35.4
	Mixed breastfeeding	66	14.7
	Other	2	0.4
	Total	449	100
Supporting person for the mother	No support	10	2.2
	Partner	341	75.9
	Mother or mother-in-law	86	19.2
	Other	12	2.7
	Total	449	100
Current perceived state of health	Good	334	74.4
	Mediocre	115	25.6
	Total	449	100
Currently on weight loss diet	No	331	73.7
	Yes	118	26.3
	Total	449	100
Physical activity performance	No physical activity	296	65.9
	Once a week	40	8.9
	Twice/three times a week	95	21.2
	Every day	18	4.0
	Total	449	100
Type of delivery	Normal delivery	280	62.4
	Instrumental delivery	58	12.9
	Caesarean section	111	24.7
	Total	449	100
Stitches	No stitches	88	19.6
	Yes, episiotomy performed	123	27.4
	Yes, tearing was mended	128	28.5
	Yes, typical of caesarean	110	24.5
	Total	449	100

**Table 3 medicina-58-00752-t003:** Correlation between EPDS and BSQ tests.

		Total BQS Test
Total EPDS Test	Correlation coefficient	0.42 *
	Sig. (bilateral)	0.001
	N	449

* Spearman’s rho correlation coefficient.

**Table 4 medicina-58-00752-t004:** Binary logistic regression study for the depression variable.

	Sig.	Exp (B)	95% C.I. for EXP (B)	
			Lower	Higher
Age	0.008	0.92	0.879	0.981
Income (less than 1000€) Ref.	0.001			
Income (between 1000 and 2000€)	0.002	0.334	0.165	0.675
Income (between 2000 and 3000€)	0.001	0.200	0.078	0.514
Income (more than 3000€)	0.777	0.846	0.265	2.695
Family type (single-parent)	0.015	5.759	1.409	23.537
Mental issues (Yes)	0.001	7.453	3.899	14.246
Perceived health (mediocre)	0.001	13.413	7.166	25.106
Stitches (No stitches received) Ref.	0.013			
Stitches (episiotomy)	0.485	0.759	0.350	1.647
Stitches (tearing)	0.003	0.288	0.128	0.647
Stitches (caesarean)	0.348	0.688	0.314	1.504
Constant	0.999	0.001		

Dependent variable: depression; Independent variables: age, marital status, employment situation, level of income, type of family, prior mental issues, level of studies, NB feeding, supporting person, perceived health, on weight loss diet, physical activity frequency, type of delivery, stitches during delivery, dissatisfaction with body image. Nagelkerke R-squared: 0.48. Hosmer–Lemeshow test (chi-squared = 25.294; *p* = 0.001).

**Table 5 medicina-58-00752-t005:** Binary logistic regression study for the dissatisfaction with body image variable.

	Sig.	Exp(B)	95% C.I. for EXP (B)	
			Lower	Higher
On a diet (yes)	0.001	4.717	2.698	8.248
Physical activity frequency (none)	0.003			
Physical activity frequency (Once a week)	0.669	0.847	0.396	1.814
Physical activity frequency (Twice/three times a week)	0.001	0.316	0.171	0.584
Physical activity frequency (every day)	0.196	0.477	0.155	1.465
Type of delivery (normal)	0.001			
Type of delivery (instrumental)	0.012	0.384	0.182	0.808
Type of delivery (caesarean)	0.012	1.897	1.152	3.123
Depression	0.001	1.119	1.074	1.166

Dependent variable: dissatisfaction with body image; Independent variables: age, marital status, employment situation, level of income, type of family, prior mental issues, level of studies, NB feeding, supporting person, perceived health, on weight loss diet, physical activity frequency, type of delivery, stitches during delivery, postpartum depression. Nagelkerke R-squared: 0.29. Hosmer–Lemeshow test (Chi-squared = 12.82; *p* = 0.11).

## Data Availability

The datasets used and/or analyzed during the current study are available from the corresponding author on reasonable request.

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
