# Peer review of "Body Image Dissatisfaction as a Risk Factor for Postpartum Depression"

_medicina, 2022, doi:10.3390/medicina58060752_

Round 1

Reviewer 1 Report

The article entitled "Body Image Dissatisfaction as a Risk Factor for Postpartum Depression" concerns an important issue and I am glad that subsequent authors take up this topic. I notice some shortcomings in the text that should be corrected.

I present my comments below:

  1. Line 27: Authors should provide the name of the tool using a capital letters for the following words that make up the name, i.e. the Edinburgh Postpartum Depression Questionnaire, and include the EPDS abbreviation in brackets for consistency.
  2. Line 29: "Body dissatisfaction was positively correlated with postpartum depression" - I suggest clarifying that it is about postpartum depression symptoms, not PPD itself
  3. Lines 119-122: “The exclusion criterion was the existence of a language barrier in terms of the Spanish language (the patient could not write, read, or speak in Spanish). In many cases, the companion or direct relative did understand Spanish, but even so, this requirement was maintained as an exclusion criterion "- are these the only exclusion criteria? What about current mental and somatic diseases? Were there any age restrictions, etc.? If they were - please describe them precisely. If not, it is a significant study limitation that should be discussed. In the following comments, I refer to the age of the respondents and their mental health.
  4. Lines 127-128: both used questionnaires - EPDS and BSQ require more detailed characterization, especially in the context of the current study. Please characterize them and provide Cronbach's alpha values ​​in the current sample.
  5. Lines 152-153: Please provide the full and correct names of the statistical tests, namely: "the Mann – Whitney U-test" instead of "Mann Whitney U" and the "Kruskal – Wallis test" instead of "Kruskal Wallis".
  6. Line 170: "(...) 18 years and the maximum being 52 years" - this is a fairly large age discrepancy in the context of the variable body (dis) satisfaction, which may be important for the results. The authors did not refer to this in either the Discussion section or the Limitation section. Firstly, I recommend the authors state how many women are in the 45-52 age range, because premenopausal symptoms may appear here, and this should be taken into account when interpreting the results. Second - why could the respondents aged 52 participate in the survey? What were the methodological assumptions? Formally, the years 15-49 are considered to be the reproductive age. If the authors took into account such age ranges, I would also recommend checking the percentage of 40-45 + mothers and including a commentary on body changes related not only to pregnancy in the discussion. Then, however, at least the age limits would have a substantive justification. Currently, the age of 52 among the participants is factually incomprehensible to me.
  7. Lines 174-175 (table 1): as I mentioned earlier, the authors do not characterize the tools at all, especially EPDS, in the context of the presented research. It is not clear what their cut-off point was or why. The average results on the EPDS scale as shown in Table 1 are below 12/13 points, i.e. the cut-off scores recommended by the authors of the original version. Perhaps the validation with Spanish samples shows the need to take into account other cut-offs. This is important, especially when the authors perform a logistic regression analysis for which the dependent variable must be dichotomous. in this case, therefore, women should be classified as depressed on the basis of the EPDS score
  8. Table 2 (History of mental health): a very large percentage of respondents (nearly 80%) declare a history of mental health. In the context of the presented study, this is a factor that may significantly distort the obtained results. In this situation, one should firstly explain/describe in detail what kind of mental health problems were experienced by the women and when they happened, and for example, compare the results in both groups. Meanwhile, the authors treat this as a PPD-increasing risk variable. Of course, it may be so, but because the methodology is not clearly presented, it is not certain that "history of mental health" does not mean that the woman developed depression just before pregnancy. Then the symptoms considered to be symptoms of PPD will de facto be a continuation of the episode that started earlier.
  9. Table 2 (Currently on weight loss diet) - there is no information on how the authors understand weight loss diet and how they formulated questions about this information. Women were examined 6 months after childbirth. It is a relatively short time from the birth of a child and is also closely related to the recommendations for exclusive breastfeeding. If a woman intentionally decides to introduce a weight loss diet (which is probably related to body dissatisfaction) in this short period of delivery, it is worth considering what came first. There is a high probability that the decision to diet is an effect/consequence of depressive symptoms, and not the other way around.
  10. Lines 204-206: This is the result to be expected. In my opinion, including such a large number of women who had mental problems in the study, in general, distorts the overall results and is not correct. Moreover, just checking if there are differences is not enough. If the authors did not include this factor in the inclusion/exclusion criteria, then in the case of such a percentage difference and obtaining a statistically significant difference in EPDS between the groups, the main analysis (logistic regression) should also be carried out separately in these groups
  11. Lines 214-216: the subtitle mentions the EPDS ("EPDS test related to child feeding method"), while the authors only refer to the result in BSQ ("Statistically significant differences were found for body image dissatisfaction (p = 215 0.02) in relation to the way the child was fed ”). There is also no information about the difference, ie in which group the results were the lowest, in which the highest. Moreover - it is not enough to provide a p-value coefficient, but also all values. As non-parametric tests based on ranks were used, their values ​​should be provided together with the results of post hoc tests when comparing more than 2 groups. This is a remark that applies to presenting the results of comparisons for each of the analyzed variables. Authors everywhere give only p-values.
  12. Line 218: "When comparing the means…" - why do authors write about means if it is nonparametric, rank-based tests? Please describe the statistical analysis correctly.
  13. Line 235: "(...) scores equal to or greater than 10 (...)" - the information on the adopted cut-off point should be described in advance and also explained. Why do the authors use the cut-off point as 10 and above. On what criteria or recommendations?
  14. Lines 238 - 240 and lines 252-255: I do not quite understand why the authors list these variables in the description of the table when they are not in the table itself, although differences in the results for these variables were previously obtained. The values ​​obtained for these variables are missing.
  15. Lines 275-277: "Our results show that single women had more PPD than married women, but divorced women had less depression than the rest" - first, what do the authors mean by "had more PPD"? Is it the frequency of the symptoms or the severity of the symptoms? Second, in the study, only 6 women out of 450 are divorced. Although the statistical program will count (almost) everything, the author is responsible for the methodological layer. I believe that it is completely unjustified to compare groups that are so unequal (even with the use of non-parametric tests). I will also repeat what I wrote earlier - on what basis do the authors indicate such results if they do not report the results of post hoc tests. Without these results, it is difficult to assess whether the mentioned differences between the groups actually exist. Post hoc tests should be performed and the results section should be supplemented with the results of these tests. Please note that non-parametric tests are based on comparing ranks, not means, and that post hoc analyzes should be used for comparisons with non-parametric tests.
  16. Lines 282-284: "In the present study, statistically significant differences were found when employment status was related to postpartum depression. Women who were unemployed or engaged in home care had higher levels of depression than women who were working "- where are the results that indicate it - it is not enough to give only the p-value, as the authors did
  17. Lines 296-297: "It is well documented that women with a history of mental illness are more likely to develop PPD" - with what mental illness? The authors should be critical of such a large percentage of women declaring mental illnesses, as it constitutes a significant limitation of the research. Especially in a situation where the mental illnesses were not controlled, when they were diagnosed, etc. In this situation the conclusion drawn by the authors seems to be obvious on the one hand and, on the other hand, invalid.
  18. Line 300: which mental illness correlates with body dissatisfaction?
  19. Lines 304-306: "In the present study, women who were on a weight loss diet had more PPD and were more dissatisfied with their body image and, ultimately, this is the reason why they started a weight loss diet" - where from the authors know that was the reason, did they collect such data in an interview or are they just speculating? if it is just a supposition it should be noted, is based on the results it should be described earlier.
  20. Lines 306-308: "Another important aspect of weight loss is physical activity, and women who did not perform physical activity at all had higher levels of PPD" - the authors conclude too arbitrarily. How do you know that women who do not undertake physical activity have higher levels of PPD, and not the other way around, i.e. women who have a high level of PPD do not undertake physical activity, which logically results from the specificity of symptoms of depression?
  21. Lines 318-319: "(...) but very few have taken into account the woman's family or social support" - I completely disagree with that. Social support and support from the immediate environment is a variable that is very often included among PPD risk factors.
  22. Line 367: the term disorder or problem sounds less stigmatizing than pathology
  23. The entire paragraph on the limitations of the study should be expanded. The authors describe only the sample bias, but to a limited extent. Meanwhile, such a large percentage of women with a previous history of mental health raises doubts, and this, in particular, should be taken into account in both the discussion and the limitation section.

Author Response

The article entitled "Body Image Dissatisfaction as a Risk Factor for Postpartum Depression" concerns an important issue and I am glad that subsequent authors take up this topic. I notice some shortcomings in the text that should be corrected.

I present my comments below:

Dear reviewer, thank you very much for all your recommendations for improvement.

Line 27: Authors should provide the name of the tool using a capital letters for the following words that make up the name, i.e. the Edinburgh Postpartum Depression Questionnaire, and include the EPDS abbreviation in brackets for consistency.

Thank you for your recommendation. The text has been amended according to your suggestion.

Line 29: "Body dissatisfaction was positively correlated with postpartum depression" - I suggest clarifying that it is about postpartum depression symptoms, not PPD itself

This suggestion has been incorporated.

Lines 119-122: “The exclusion criterion was the existence of a language barrier in terms of the Spanish language (the patient could not write, read, or speak in Spanish). In many cases, the companion or direct relative did understand Spanish, but even so, this requirement was maintained as an exclusion criterion "- are these the only exclusion criteria? What about current mental and somatic diseases? Were there any age restrictions, etc.? If they were - please describe them precisely. If not, it is a significant study limitation that should be discussed. In the following comments, I refer to the age of the respondents and their mental health.

Age range was not an exclusion criterion. All women who gave birth were included. In our sample, the youngest woman was 18 years old and the oldest, 52 years old.

Regarding mental illness, they were asked whether they had a history of mental illness. In the discussion, between lines 297 and 300, there is an explanation that says that there are previous studies that state that people with a history of mental problems are more likely to have postpartum depression and more dissatisfaction with their body image. Therefore, in our study, having a history of mental health was not an exclusion criterion.

Lines 127-128: both used questionnaires - EPDS and BSQ require more detailed characterization, especially in the context of the current study. Please characterize them and provide Cronbach's alpha values ​​in the current sample.

The characteristics of the scales were explained and Cronbach's alpha data for our sample were provided.

Lines 152-153: Please provide the full and correct names of the statistical tests, namely: "the Mann – Whitney U-test" instead of "Mann Whitney U" and the "Kruskal – Wallis test" instead of "Kruskal Wallis".

The full name of the test is now provided.

Line 170: "(...) 18 years and the maximum being 52 years" - this is a fairly large age discrepancy in the context of the variable body (dis) satisfaction, which may be important for the results. The authors did not refer to this in either the Discussion section or the Limitation section. Firstly, I recommend the authors state how many women are in the 45-52 age range, because premenopausal symptoms may appear here, and this should be taken into account when interpreting the results. Second - why could the respondents aged 52 participate in the survey? What were the methodological assumptions? Formally, the years 15-49 are considered to be the reproductive age. If the authors took into account such age ranges, I would also recommend checking the percentage of 40-45 + mothers and including a commentary on body changes related not only to pregnancy in the discussion. Then, however, at least the age limits would have a substantive justification. Currently, the age of 52 among the participants is factually incomprehensible to me.

There was only one woman aged 52 in our sample, representing 0.2% of the sample. If we eliminated this record, the age range would be between 18 and 44 years. To avoid confusion and having to introduce more variables, we opted for eliminating this record. As a consequence, the statistical study had to be recalculated, modifying all those data that were altered in the paper. In addition, age over 44 years was included as an exclusion criterion. We are aware that there are very few women of childbearing age over the age of 44.

Lines 174-175 (table 1): as I mentioned earlier, the authors do not characterize the tools at all, especially EPDS, in the context of the presented research. It is not clear what their cut-off point was or why. The average results on the EPDS scale as shown in Table 1 are below 12/13 points, i.e. the cut-off scores recommended by the authors of the original version. Perhaps the validation with Spanish samples shows the need to take into account other cut-offs. This is important, especially when the authors perform a logistic regression analysis for which the dependent variable must be dichotomous. in this case, therefore, women should be classified as depressed on the basis of the EPDS score

The characteristics of the scale have been explained in the manuscript. In the Spanish validation, the cut-off point was considered to be 10 points. Thus, in order to code the variable as dichotomous for the regression study, people who scored 10 points or more were considered depressive.

Table 2 (History of mental health): a very large percentage of respondents (nearly 80%) declare a history of mental health. In the context of the presented study, this is a factor that may significantly distort the obtained results. In this situation, one should firstly explain/describe in detail what kind of mental health problems were experienced by the women and when they happened, and for example, compare the results in both groups. Meanwhile, the authors treat this as a PPD-increasing risk variable. Of course, it may be so, but because the methodology is not clearly presented, it is not certain that "history of mental health" does not mean that the woman developed depression just before pregnancy. Then the symptoms considered to be symptoms of PPD will de facto be a continuation of the episode that started earlier.

Indeed, the question asked in the questionnaire was: Before pregnancy, did you have any problems with depression, anxiety, or eating disorders such as anorexia or bulimia? And almost 80% answered no. The intention was to see that people who reported having had such mental health problems before pregnancy were more dissatisfied with their body image and had more depression postpartum than women who had not had mental health problems. The mean score on the EPDS test for those without mental health problems was 7.44, versus 13.42 for those who reported mental health problems (p=0.001). The mean score on the BSQ test for people who reported not having had mental health problems was 19.48, versus 24.97 for people who reported having had mental health problems (p=0.001).

Table 2 (Currently on weight loss diet) - there is no information on how the authors understand weight loss diet and how they formulated questions about this information. Women were examined 6 months after childbirth. It is a relatively short time from the birth of a child and is also closely related to the recommendations for exclusive breastfeeding. If a woman intentionally decides to introduce a weight loss diet (which is probably related to body dissatisfaction) in this short period of delivery, it is worth considering what came first. There is a high probability that the decision to diet is an effect/consequence of depressive symptoms, and not the other way around.

The question was formulated as follows: Are you currently on a weight loss diet with the sole aim of losing weight?

The mean score on the EPDS test for women who were not on a weight loss diet was 8.28, while the mean score for dieters was 9.69 (p= 0.03). For the BSQ test, the mean score of the non-dieters was 19.02, while dieters had a mean score of 25.00 (p=0.001).

Lines 204-206: This is the result to be expected. In my opinion, including such a large number of women who had mental problems in the study, in general, distorts the overall results and is not correct. Moreover, just checking if there are differences is not enough. If the authors did not include this factor in the inclusion/exclusion criteria, then in the case of such a percentage difference and obtaining a statistically significant difference in EPDS between the groups, the main analysis (logistic regression) should also be carried out separately in these groups

There was a transcription error and the data were reversed. Actually, there were only 20% of women who had a history of mental health problems compared to 80% who did not. The data in the table has been changed. The error only appeared in table 2 and has been corrected. The entire statistical study was reviewed and the regression analysis was found to be correct.  However, this condition will be explained in the limitations section.

Lines 214-216: the subtitle mentions the EPDS ("EPDS test related to child feeding method"), while the authors only refer to the result in BSQ ("Statistically significant differences were found for body image dissatisfaction (p = 215 0.02) in relation to the way the child was fed ”). There is also no information about the difference, ie in which group the results were the lowest, in which the highest. Moreover - it is not enough to provide a p-value coefficient, but also all values. As non-parametric tests based on ranks were used, their values ​​should be provided together with the results of post hoc tests when comparing more than 2 groups. This is a remark that applies to presenting the results of comparisons for each of the analyzed variables. Authors everywhere give only p-values.

Depression was the intended word, as indicated in the subtitle. By mistake, body image dissatisfaction was written. It has been corrected in the paper.

Line 218: "When comparing the means…" - why do authors write about means if it is nonparametric, rank-based tests? Please describe the statistical analysis correctly.

Thank you very much, as you are right indeed. These are non-parametric tests, so it is not possible to compare means. Statistically significant differences can be established between variables but not between means. It was an editorial error that has been corrected in the paper.

Line 235: "(...) scores equal to or greater than 10 (...)" - the information on the adopted cut-off point should be described in advance and also explained. Why do the authors use the cut-off point as 10 and above. On what criteria or recommendations?

This cut-off point was established by the authors and developers of the scale.

Cox, J. L., Holden, J. M., & Sagovsky, R. (1987). Detection of postnatal depression. Development of the 10-item Edinburgh Postnatal Depression Scale. The British journal of psychiatry : the journal of mental science150, 782–786. https://doi.org/10.1192/bjp.150.6.782

It has been explained and described in the document.

Lines 238 - 240 and lines 252-255: I do not quite understand why the authors list these variables in the description of the table when they are not in the table itself, although differences in the results for these variables were previously obtained. The values ​​obtained for these variables are missing.

In the binary logistic regression analysis, all the variables that showed statistically significant differences in the bivariate analysis were considered independent variables. They are necessary to perform the regression, but the model refines all the variables in multiple steps so that in the final step only those that are included in the table are expressed. Therefore, no data is missing, the model simply left only those that were calculated as significant in the last step.

Lines 275-277: "Our results show that single women had more PPD than married women, but divorced women had less depression than the rest" - first, what do the authors mean by "had more PPD"? Is it the frequency of the symptoms or the severity of the symptoms? Second, in the study, only 6 women out of 450 are divorced. Although the statistical program will count (almost) everything, the author is responsible for the methodological layer. I believe that it is completely unjustified to compare groups that are so unequal (even with the use of non-parametric tests). I will also repeat what I wrote earlier - on what basis do the authors indicate such results if they do not report the results of post hoc tests. Without these results, it is difficult to assess whether the mentioned differences between the groups actually exist. Post hoc tests should be performed and the results section should be supplemented with the results of these tests. Please note that non-parametric tests are based on comparing ranks, not means, and that post hoc analyzes should be used for comparisons with non-parametric tests.

Having more than 10 points on the scale indicates that the woman is very likely to have postpartum depression, but this scale cannot give us the severity of the symptoms. Therefore, when the authors say that the woman has more PPD they mean that the score on the scale is higher and, therefore, they are more likely to have PPD. The sentence has been rewritten.

Thank you very much; you are right to note that the number of divorced women is very small in relation to the total sample. This explanation has been added in the text, in the discussion section.

It has been included in the post hoc analysis for all results.

Lines 282-284: "In the present study, statistically significant differences were found when employment status was related to postpartum depression. Women who were unemployed or engaged in home care had higher levels of depression than women who were working "- where are the results that indicate it - it is not enough to give only the p-value, as the authors did

All results data have been added.

Lines 296-297: "It is well documented that women with a history of mental illness are more likely to develop PPD" - with what mental illness? The authors should be critical of such a large percentage of women declaring mental illnesses, as it constitutes a significant limitation of the research. Especially in a situation where the mental illnesses were not controlled, when they were diagnosed, etc. In this situation the conclusion drawn by the authors seems to be obvious on the one hand and, on the other hand, invalid.

The women were asked if they had a history of mental health problems diagnosed by a doctor. They were asked to say which one. All those who answered yes reported that these illnesses were anxiety and/or depression. The studies that were consulted established that women with pre-birth anxiety and/or depression are more likely to develop postpartum depression. This has been explained in the text.

Line 300: which mental illness correlates with body dissatisfaction?

It is anxiety and depression. This has been specified in the text.

Lines 304-306: "In the present study, women who were on a weight loss diet had more PPD and were more dissatisfied with their body image and, ultimately, this is the reason why they started a weight loss diet" - where from the authors know that was the reason, did they collect such data in an interview or are they just speculating? if it is just a supposition it should be noted, is based on the results it should be described earlier.

The women were asked whether they were currently on a weight loss diet because they felt fat and, therefore, dissatisfied with their bodies. The question was clear and left no room for speculation.

The mean body image dissatisfaction was analysed together with the depression scale in both groups (those on a weight loss diet and those not on a weight loss diet) and it was found that women who were on a weight loss diet were more dissatisfied with their body image and scored higher on the EPDS.

This has been detailed in the results.

Lines 306-308: "Another important aspect of weight loss is physical activity, and women who did not perform physical activity at all had higher levels of PPD" - the authors conclude too arbitrarily. How do you know that women who do not undertake physical activity have higher levels of PPD, and not the other way around, i.e. women who have a high level of PPD do not undertake physical activity, which logically results from the specificity of symptoms of depression?

As mentioned above, the score on the EPDS scale does not correlate with the severity of symptoms. We could see that women who are not physically active have higher EPDS scores. This means that they are more likely to be suffering from postpartum depression. In our research, we compared being physically active with the EPDS scale score and it was found that people who were not physically active had higher scores on the EPDS scale. This has been clarified in the text to avoid confusion.

Lines 318-319: "(...) but very few have taken into account the woman's family or social support" - I completely disagree with that. Social support and support from the immediate environment is a variable that is very often included among PPD risk factors.

Thank you for your comment; we fully agree.

Line 367: the term disorder or problem sounds less stigmatizing than pathology

Thank you for your suggestion; the term has been changed to ‘disorder’ as suggested.

The entire paragraph on the limitations of the study should be expanded. The authors describe only the sample bias, but to a limited extent. Meanwhile, such a large percentage of women with a previous history of mental health raises doubts, and this, in particular, should be taken into account in both the discussion and the limitation section.

As mentioned above, there was a transcription error. After correction, the percentage of women with a history of anxiety and/or depression is only 20.3%. This figure has been mentioned in the limitations section.

Reviewer 2 Report

Thank you for giving me the opportunity to review this manuscript.

I think it is necessary to revise the manuscript before publication.

1) Please explain the study design by using PECO format in the abstract and the method.

2) Pleae describe the relevant dates, including periods of recruitment, exposure, follow-up, and data collection in the method section.

3)Please explain the Edinburgh Postpartum Depression Scale (EPDS) and the Body Shape Questionnaire (BSQ) more in detail. How long does it take to finish using this scale? What is the maximum and minimum score? What is the normal range of this score?

4) Please describe any efforts to address potential sources of bias. Please explain how missing data was addressed and analyzed.

5) Please move the paragraph of "2.6 Ethical aspects " to the section below the conclusion, because this is not the method.

6) Please define what is body image dissatisfaction positive group (exposure) and  body image dissatisfaction positive group (control). I think it is necessary to show the baseline characteristics of these groups in the result section. Please indicate number of participants with missing data for each variable of interest.

7) Please describe unadjusted estimates (Model 1) and, if applicable, confounder-adjusted estimates and their precision (Model 2, 3, 4) (eg, 95% confidence interval). Make clear which confounders were adjusted for and why they were included. I think that this study cannot exclude the impact of confounders on the relationship between the Edinburgh Postpartum Depression and the Body image dissatisfaction. 

8)   I cannot understand why the following method was selected for the statistical analysis; "In the bivariate statistical analysis, to relate quantitative variables and after the Kormogorov-Smirnov normality test, non-parametric tests were used: Mann Whitney U to compare two independent variables and Kruskal Wallis for more than two.".  I think what the authors want to show in this study is obscure. Please define what is the main outcome and what is the confounders in the method section. I think that the authors mixed up the outcomes, the exposures, and the confounders (age, marital status, employment situation, level of income, type of family, prior mental issues, level of studies, NB feeding, supporting person, perceived health, on weight loss diet, physical activity frequency, type of delivery, stitches during delivery??). Please modify these sentences. Furthermore, why wasn't propensity score matching selected for statistical analysis although there were so many confounders?? I think that this study cannot controll for confounders. Please explain what is the predictors, potential confounders and effect modifiers, and how to controll for these factors by using the statistical analysis.

9) Give a cautious overall interpretation of results considering objectives, limitations, multiplicity of analyses, results from similar studies, and other relevant evidence. Please re-write the discussion section by modifying the sentences according to the points above. Samely, please change the sentences of "The results obtained in this study reveal that dissatisfaction with body image directly affects PPD, favouring its appearance; therefore, it is necessary to promote a positive attitude towards body image and provide women with the necessary tools to improve their body image in the postpartum period, thus reducing the risk of suffering PPD." after reanalyzing the results.

10) English should be proofread by English experts.

I think that it is necessary to revise the manuscript.

Author Response

Thank you for giving me the opportunity to review this manuscript.

Dear reviewer, thank you very much for all your recommendations for improvement.

I think it is necessary to revise the manuscript before publication.

1) Please explain the study design by using PECO format in the abstract and the method.

Thank you very much for your recommendation. PICO format is incorporated into section 2.1. ‘Study design’.

2) Pleae describe the relevant dates, including periods of recruitment, exposure, follow-up, and data collection in the method section.

Section 2.1 ‘Population and sample’ states that the sample was collected from births occurring between 1 July 2020 and 20 June 2021.

Section 2.4 ‘Data collect’ explains that the data collection period was from 1 January to 31 December 2021.

3) Please explain the Edinburgh Postpartum Depression Scale (EPDS) and the Body Shape Questionnaire (BSQ) more in detail. How long does it take to finish using this scale? What is the maximum and minimum score? What is the normal range of this score?

It has been explained in more detail.

4) Please describe any efforts to address potential sources of bias. Please explain how missing data was addressed and analyzed.

There were no missing data. The online questionnaire was set up in such a way that all questions were mandatory. Therefore, if the respondent tried to leave a blank answer, the system would not let them continue.

5) Please move the paragraph of "2.6 Ethical aspects " to the section below the conclusion, because this is not the method.

In line with your proposal, that paragraph has been moved to the recommended location.

6) Please define what is body image dissatisfaction positive group (exposure) and  body image dissatisfaction positive group (control). I think it is necessary to show the baseline characteristics of these groups in the result section. Please indicate number of participants with missing data for each variable of interest.

How the two groups were chosen was added in the results section, specifically in the multivariate analysis. This is important for regression studies.

7) Please describe unadjusted estimates (Model 1) and, if applicable, confounder-adjusted estimates and their precision (Model 2, 3, 4) (eg, 95% confidence interval). Make clear which confounders were adjusted for and why they were included. I think that this study cannot exclude the impact of confounders on the relationship between the Edinburgh Postpartum Depression and the Body image dissatisfaction.

As you may be aware, in binary logistic regression there are several ways to measure goodness of fit, so-called tests based on patterns of covariates, such as the test based on the D-deviance and Pearson's Chi-squared statistic, and tests based on estimated probabilities, such as Hosmer's and Lemeshow's; as can be seen, both appear in all the tables at the bottom.

On the other hand, all those variables that were statistically significant in the bivariate analysis should be included as dependent variables, together with those that the researcher considers important for their epidemiological approach. Thus, the variables included in the model appear at the bottom of the tables, leaving in the body of the table those that the model selects, with a significance of p < 0.05. Likewise, logistic regression models allow for adjustment for confounding by adjusting the ORs for all other variables (optimal confounding adjustment).

8)   I cannot understand why the following method was selected for the statistical analysis; "In the bivariate statistical analysis, to relate quantitative variables and after the Kormogorov-Smirnov normality test, non-parametric tests were used: Mann Whitney U to compare two independent variables and Kruskal Wallis for more than two.".  I think what the authors want to show in this study is obscure. Please define what is the main outcome and what is the confounders in the method section. I think that the authors mixed up the outcomes, the exposures, and the confounders (age, marital status, employment situation, level of income, type of family, prior mental issues, level of studies, NB feeding, supporting person, perceived health, on weight loss diet, physical activity frequency, type of delivery, stitches during delivery??). Please modify these sentences. Furthermore, why wasn't propensity score matching selected for statistical analysis although there were so many confounders?? I think that this study cannot controll for confounders. Please explain what is the predictors, potential confounders and effect modifiers, and how to controll for these factors by using the statistical analysis.

The authors consider that the method used and described is the ideal method for comparing means of NON-NORMAL quantitative variables, presenting the data in the clearest possible and most correct methodological way that we consider.

In relation to your comment on the confounding factors, continuing with the answer to the previous point, they are minimised with the use of logistic regression.

In response to your request, the sentences have been modified.

The predictor variables and the weight of each one of them (OR) in the adjusted model can be seen in the tables below.

9) Give a cautious overall interpretation of results considering objectives, limitations, multiplicity of analyses, results from similar studies, and other relevant evidence. Please re-write the discussion section by modifying the sentences according to the points above. Samely, please change the sentences of "The results obtained in this study reveal that dissatisfaction with body image directly affects PPD, favouring its appearance; therefore, it is necessary to promote a positive attitude towards body image and provide women with the necessary tools to improve their body image in the postpartum period, thus reducing the risk of suffering PPD." after reanalyzing the results.

The authors consider, on the basis of the above analysis, that the meaning of the sentence is correct.

10) English should be proofread by English experts.

The manuscript has been reviewed by an English specialist and a native speaker.

I think that it is necessary to revise the manuscript.

Thank you very much for all your comments.

Round 2

Reviewer 2 Report

Thank you for revising the manuscript. 

It is necessary to modify some sentence. I think that the following description is wrong.

"The study design was based on the following research question: Is there a relationship between body image (dis)satisfaction and the likelihood of developing postpartum depression in post-  partum women? According to the PICO format, the (P)opulation is postpartum women,  the (I)ntervention of the study is to assess body image dissatisfaction to (C)ompare it with  the likelihood of developing postpartum depression according to the (O)utcomes based  on marital status, employment status, socio-economic status, educational level, having a mental health history, being on a weight loss diet, physical activity, child feeding method,  having support from others, self-perceived health status, and the type of delivery."

Because this is the cross-sectional study (observational study), Please describe the study design by PECO format. P means the population. E means Exposure. C means control. O means outcomes.  According to the title, I think that  E is the Body Image Dissatisfaction group (specific scores of BSQ?), while C is the body image satiscaction group (specific scores of BSQ?) in this study. The outcome should be scores of EPDS (the primary outcome?). Furthermore, I think that the covariates may be the followings; marital status, employment status, socio-economic status, educational level, having a mental health history, being on a weight loss mdiet, physical activity, child feeding method,  having support from others, self-perceived health status, and the type of delivery.

Please modify the sentences to describe the study design correctly.

I cannot accept this manuscript for publication in the current form.

Author Response

Thank you for revising the manuscript. 

It is necessary to modify some sentence. I think that the following description is wrong.

 "The study design was based on the following research question: Is there a relationship between body image (dis)satisfaction and the likelihood of developing postpartum depression in postpartum women? According to the PICO format, the (P)opulation is postpartum women,  the (I)ntervention of the study is to assess body image dissatisfaction to (C)ompare it with  the likelihood of developing postpartum depression according to the (O)utcomes based  on marital status, employment status, socio-economic status, educational level, having a mental health history, being on a weight loss diet, physical activity, child feeding method,  having support from others, self-perceived health status, and the type of delivery."

 Because this is the cross-sectional study (observational study), Please describe the study design by PECO format. P means the population. E means Exposure. C means control. O means outcomes.  According to the title, I think that  E is the Body Image Dissatisfaction group (specific scores of BSQ?), while C is the body image satiscaction group (specific scores of BSQ?) in this study. The outcome should be scores of EPDS (the primary outcome?). Furthermore, I think that the covariates may be the followings; marital status, employment status, socio-economic status, educational level, having a mental health history, being on a weight loss mdiet, physical activity, child feeding method,  having support from others, self-perceived health status, and the type of delivery.

 Please modify the sentences to describe the study design correctly.

I cannot accept this manuscript for publication in the current form.

Thank you very much for your prompt response, for your suggestion and your time. The authors believe that the implementation of the PECO format is very appropriate, so we agree to its modification in the text.

This manuscript is a resubmission of an earlier submission. The following is a list of the peer review reports and author responses from that submission.